## Perspective

machine learning; vibrational spectroscopy; O-PTIR; super resolution; organelle mapping

**Corresponding author:**
Joachim Heberle;
Email: jheberle@zedat.fu-berlin.de

# Infrared nanoscopy for subcellular chemical imaging

Katerina Kanevche[1] [iD], David Joll Burr[2,3], Janina Drauschke[2], Jacek Kozuch[4], Carlos Baiz[5], Andreas Elsaesser[2] and Joachim Heberle[4] [iD]

[1]Department of Chemistry, Princeton University, Princeton, NJ, USA; [2]Department of Physics, Experimental Biophysics and Space Sciences, Freie Universität Berlin, Berlin, Germany; [3]Institute for Biology – Microbiology, Freie Universität Berlin, Berlin, Germany; [4]Department of Physics, Experimental Molecular Biophysics, Freie Universität Berlin, Berlin, Germany and [5]Department of Chemistry, University of Texas at Austin, Austin, TX, USA

## Abstract

Infrared (IR) nanoscopy represents a collection of imaging and spectroscopy techniques capable of resolving IR absorption on the nanometer scale. Chemical specificity is leveraged from vibrational spectroscopy, while light–matter interactions are detected by observing perturbations in the optical near field with an atomic force microscopy probe. Therefore, imaging is wavelength independent and has a spatial resolution on the nanometer scale, well beyond the classical diffraction limit. In this perspective, we outline the recent biological applications of scattering type scanning near-field optical microscopy and nanoscale Fourier-transform IR spectroscopy. These techniques are uniquely suited to resolving subcellular ultrastructure from a variety of cell types, as well as studying biological processes such as metabolic activity on the single-cell level. Furthermore, this review describes recent technical advances in IR nanoscopy, and emerging machine learning supported approaches to sampling, signal enhancement, and data processing. This emphasizes that label-free IR nanoscopy holds significant potential for ongoing and future biological applications.

## Introduction

Studying biomolecular processes in cells frequently relies on the ability to resolve chemical composition on a subcellular level, and thus requires nanoscale spatial resolution. Some established routes to subcellular imaging include super-resolution fluorescence microscopy (Sahl *et al.*, 2017), or electron microscopy (EM) (Nogales and Mahamid, 2024). However, fluorescence-based microscopy requires labeling of the biomolecule of interest with a fluorescent tag. This carries certain drawbacks including prior knowledge of the sample composition, the development and availability of suitable fluorescent tag molecules, and the inclusion of fluorophores potentially introducing toxicity or perturbing the native state of the biological system being interrogated. Cryo-EM provides unprecedented spatial resolution down to the Ångstrom level and the cryogenic temperatures can mitigate sample damage from high-energy electrons. However, this technique is very costly to operate, involves complex computationally expensive data processing algorithms, and lacks direct information on the chemical composition.

Nondestructive techniques that are inherently sensitive to chemical composition, while allowing for mapping on the nanometer scale, are a promising alternative for cellular bioimaging. In the realm of tip-enhanced microscopy and spectroscopy, methodologies that combine scanning probe microscopy (SPM) and vibrational spectroscopy are particularly attractive. Atomic force microscopy (AFM), a type of SPM, is a surface-sensitive technique frequently employed for the characterization of soft matter and biological samples, which yields spatial resolution on the nanometer level (Allison *et al.*, 2010; Müller and Dufrêne, 2011). Combining this method with vibrational spectroscopy overcomes the diffraction limit by accessing the light–matter interaction in the optical near field via the scanning probe, thus, the molecular vibrations can be resolved with the spatial resolution of AFM.

Scattering type scanning near-field optical microscopy (sSNOM) (Figure 1) is one example that utilizes an AFM tip as an antenna to localize electromagnetic radiation to the immediate proximity of the tip. This approach provides routine spatial resolutions of 20 nm. sSNOM images are recorded using tunable laser sources, with the majority of its applications in the mid-IR and THz spectral range. When employing broadband laser sources, the spectra are recorded via a Fourier transform-based spectrometer, where the sample arm of the interferometer entails the AFM tip–sample interaction. This technique is known as nanometer Fourier transform infrared (nanoFTIR) (Huth *et al.*, 2012). Both approaches are encompassed under the term IR nanoscopy. With the emergence of novel IR laser sources (Faist *et al.*, 1994; Keilmann and Amarie, 2012) and the development of a sophisticated detection scheme that allows for the extraction of optical

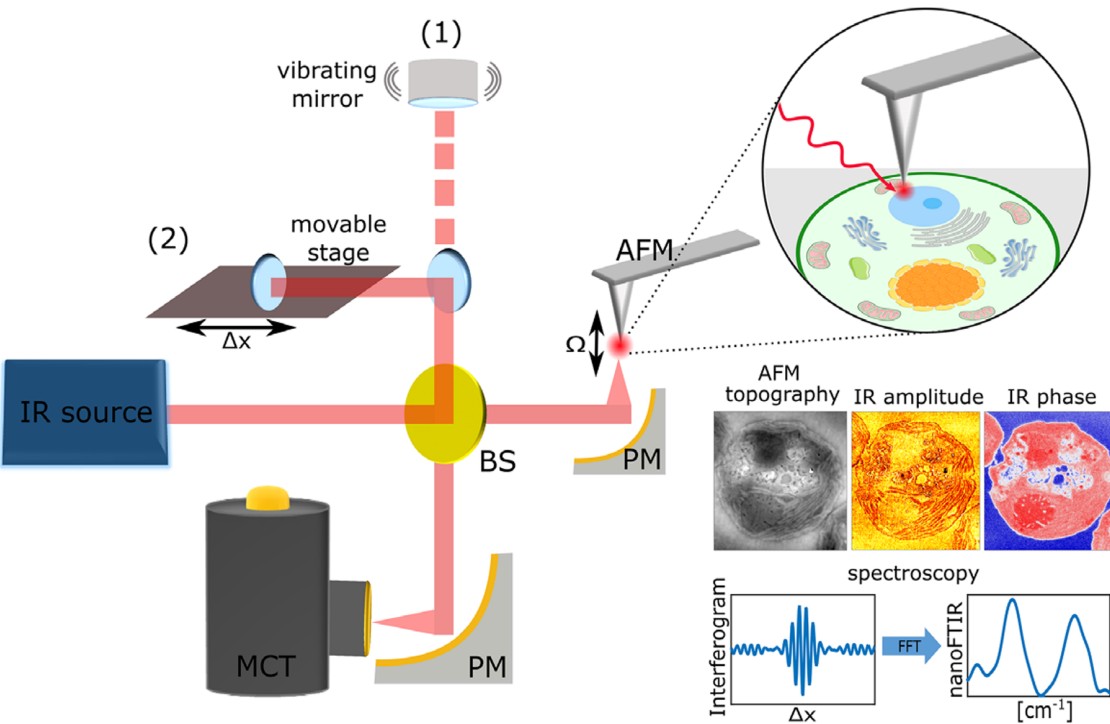

**Figure 1. Technical overview of IR nanoscopy.** Typical sSNOM (1) and nanoFTIR (2) setup based on an asymmetric Michelson interferometer. An IR light source is focused on an AFM tip, oscillating at frequency Ω, via parabolic mirror (PM). The backscattered light is collected via the same pathway, recombined with the reference beam at the beam splitter (BS) and focused on a mercury-cadmium-telluride (MCT) detector. This technique allows for recording the AFM topography, IR phase, and amplitude, which reveal the absorption and reflection of the specimen. Fourier transformation of the interferograms yields nanoFTIR spectra, thus capturing IR absorption across broad spectral range. Figure adapted from (Kanevche et al., 2021).

material properties with nm resolution (Ocelic *et al.*, 2006), the applications of sSNOM and nanoFTIR have dramatically increased across fields. One complementary technique to sSNOM is AFM-IR, where the imaging contrast originates from photothermal-induced resonance in the sample (Dazzi *et al.*, 2012). A recent review provides an excellent overview of developments and applications of nanoscopy (Hillenbrand *et al.*, 2025), while the key features of AFM-based mid-IR nanoscopy techniques are summarized in Table 1.

Applying IR nanoscopy to soft matter and biological specimens presents additional challenges regarding adequate sample preparation and generally weaker signals. Successful nanoscale imaging and spectroscopy has been demonstrated on a variety of organic samples such as membrane proteins in lipids (Ballout *et al.*, 2011),

individual protein complexes (Amenabar *et al.*, 2013; Berweger *et al.*, 2013), lipids (Kästner *et al.*, 2018), polypeptides and amyloids (Fernandes *et al.*, 2022; Paul *et al.*, 2023), and plant cell walls (Keplinger *et al.*, 2014; Veber *et al.*, 2023) to name just a few.

Here, we focus on the various approaches taken for studying cellular structure and function, such as metabolic activity, using sSNOM and nanoFTIR in the mid-IR spectral range. We argue that IR nanoscopy can be utilized for correlative imaging in concert with other imaging techniques, as well as offering an alternative route for cellular bioimaging, and thus we highlight the potential of this technology. Specifically, we emphasize the developments in sub-cellular IR chemical imaging, outline applications utilizing nano-scale IR biomarker monitoring, and provide an overview of emerging technical advances, machine learning supported

**Table 1.** Summary of AFM-based mid-IR nanoscopy techniques

| Technique | Contrast mechanism | Light source | Spectral range | Resolution | Advantages |
|---|---|---|---|---|---|
| sSNOM | Elastic scattering | Tunable laser (QCL, OPO) | QCL: ~800–2500 cm$^{-1}$<br>OPO: 550–7000 cm$^{-1}$ | Record ~5 nm<br>Typical <20 nm | Detection of optical properties of biomaterials (scattered amplitude and phase).<br>Laser sources with high power output. |
| NanoFTIR | Elastic scattering | Broadband radiation (DFG laser, synchrotron) | DFG instantaneous ~950–1900 cm$^{-1}$<br>Synchrotron instantaneous: 600–4000 cm$^{-1}$ | Typical <20 nm | Detection of optical properties of biomaterials (scattered amplitude and phase). Can accommodate ultrafast spectroscopy. |
| AFM-IR | Photothermal expansion | Tunable laser (QCL, OPO) | QCL: ~800–2500 cm$^{-1}$<br>OPO: 550–7000 cm$^{-1}$ | Typical <20 nm | Laser sources with high-power output.<br>Simpler setup (no IR detector, no interferometric detection). |

The listed techniques can in principle operate in the visible and THz regime, as discussed in the review previously (Hillenbrand *et al.*, 2025).

sampling, and postprocessing algorithms. We conclude with an outlook on how this technology may develop in the future.

## Subcellular IR imaging

The ability of IR-nanoscopic techniques to move beyond the diffraction limit and resolve chemical composition on the nm level has resulted in community interest expanding from biological soft matter characterization to the study of whole model cellular organisms. Single-cell, label-free chemical imaging was achieved using sSNOM as early as 2016, wherein intracellular chemical mapping was performed on single human red blood cells (Amrania *et al.*, 2016). This provided a significant milestone toward applying IR nanoscopy in biomedical contexts.

As sSNOM is a surface-sensitive technique, subcellular IR nanoscopy requires a sample preparation protocol that provides the AFM probe access to the subcellular structure. To maintain high-quality sSNOM signal, it is crucial to avoid any tip damage or contamination through interaction with the samples. Operating the tip in tapping mode, minimizes these common challenges when using AFM for biological samples. To optimize the sSNOM signal, it is recommended to deposit samples on an atomically flat metallic substrate. Finally, it is best practice to use flat samples with a thickness similar to the IR near-field penetration depth. To achieve these requirements, cells or biological tissue can be thin sectioned using ultramicrotomy (Herrmann, 2021), a procedure frequently employed in EM, and deposited on silicon or template-stripped gold (e.g., Horstmann *et al.*, 2012, and references therein). While robust samples, such as woody plant cells, can be sliced in this manner directly (Keplinger *et al.*, 2014), the imaging of

ultrastructure or subcellular details in soft-matter samples typically requires cryopreservation or resin embedding prior to sectioning. Resin embedding poses less logistical challenges than cryosectioning and results in samples that can be indefinitely maintained at room temperature, with no measurement-induced loss in quality. However, it must be ensured that the IR features of the embedding resin itself do not obscure spectroscopic details of interest, and as chemical fixation or freeze substitution are necessary, this is not suitable for all sample types.

To our knowledge, the first instance of subcellular chemical imaging demonstrated melanin pigments within retina cells of zebrafish (Figure 2*a*) (Stanciu *et al.*, 2017). This initial application of subcellular sSNOM utilized a laser in the visible range (638 nm), but broader chemical information can be derived by probing in the mid-IR range. The first report of subcellular IR nanoscopy demonstrated the local absorption of subcellular membrane-bound organelles within the green alga *Chlamydomonas reinhardtii* (Figure 2*b*) (Kanevche *et al.*, 2021). Absorption imaging at different wavenumbers (i.e., 1655 and 1540 cm$^{-1}$) corresponded to unique cellular features. High absorption in the amide I and amide II regions demonstrated elevated protein concentrations within the pyrenoid, an organelle tightly packed with the enzyme Rubisco. Biomolecular condensates, such as nuclear bodies within the nucleus, were uniquely visible in the IR, but not via AFM topographic imaging. Furthermore, the remarkable resolving power of sSNOM revealed IR signatures of ultrafine structures within the cellular flagella which were as small as 30 nm as well as of the sub-20-nm-sized outer cell membrane. Being able to chemically resolve the detailed internal structure of *C. reinhardtii* on the nanoscale serves as a proof of concept for future applications that

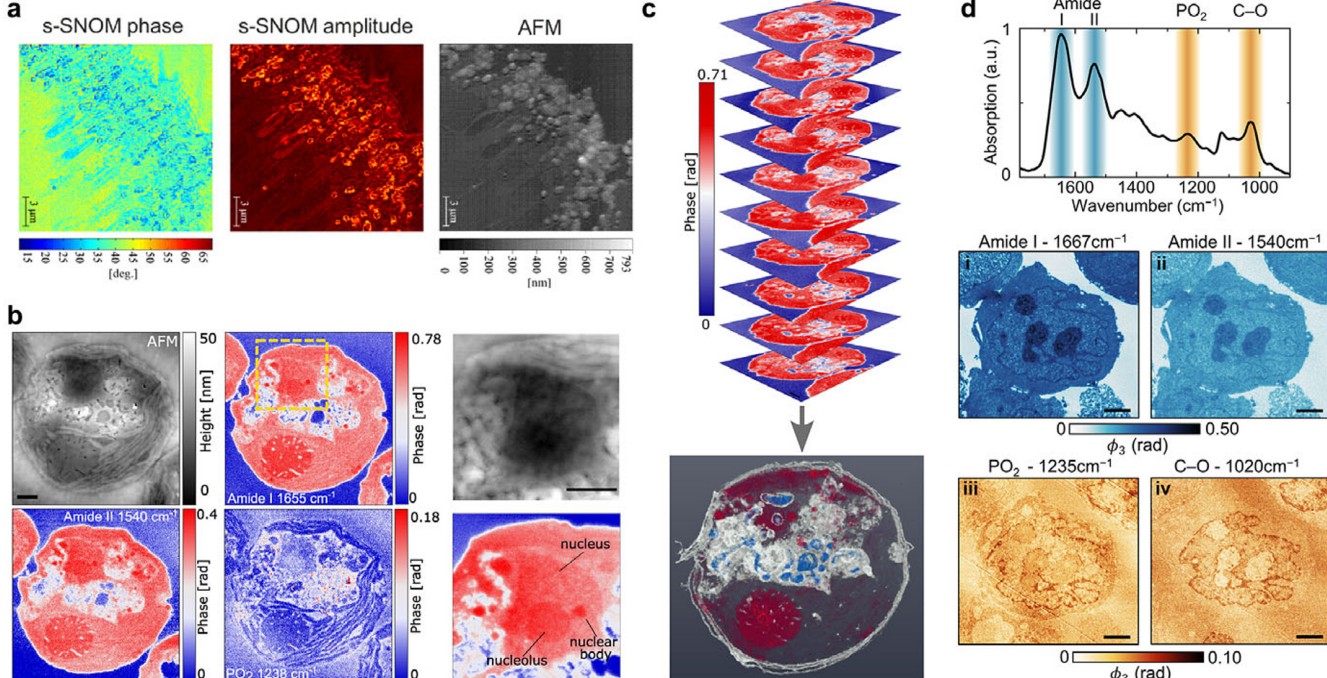

**Figure 2. Subcellular nanoscopy examples**. A. AFM and sSNOM imaging (at 638 nm) of thin-sectioned zebrafish (Danio rerio) retina. B. AFM and sSNOM imaging (at wavelengths of interest) of thin-sectioned green algae (C. reinhardtii) cells, primarily revealing the protein distribution within the cell. C. sSNOM tomography constructed from ten sequential images of C. reinhardtii, imaged at 1655 cm$^{-1}$. Video demonstrating the 3D structure of the green algae cell is available here: https://static-content.springer.com/esm/art%3A10.1038%2Fs42003-021-02876-7/MediaObjects/42003_2021_2876_MOESM4_ESM.mpg. D. Absorption spectrum of myeloma thin sections and sSNOM images at four wavelengths of interest, revealing the subcellular components. Figure adapted with permission from Stanciu et al., 2017 © Optical Society of America, and (Greaves et al., 2023; Kanevche et al., 2021).

seek to answer outstanding questions in biology such as relating cell function to compartmentalization or how biochemical processes rely on the precise location of metabolites with cells.

Performing IR-nanoscopic imaging on cellular thin sections can provide unprecedented subcellular chemical imaging detail in lateral dimensions, however, further information can be accessed in the z dimension. Other imaging techniques, including AFM (Chen *et al.*, 2005) and serial block-face scanning EM (Denk and Horstmann, 2004), have successfully employed a tomographic approach, sequentially examining consecutive cell sections to compile 3D cellular information. The first instance of IR cell tomography imaged the amide I absorption (1655 $cm^{-1}$) of a single *C. reinhardtii* cell in 10 consecutive, 100-nm-thick sections (Kanevche *et al.*, 2021). The resulting 2D images were superimposed, revealing the 3D amide I absorption of *C. reinhardtii* (Figure 2c).

Several contemporary studies have applied subcellular IR microscopy and sSNOM to investigate human disease and pathology (Freitas *et al.*, 2021; Greaves *et al.*, 2023, 2024a,b; Keogan *et al.*, 2025). Intracellular IR imaging has been performed on human myeloma cells, identifying both the local protein content and isolated nucleic acid signatures through visualization at wavelengths corresponding to the amide regions, and the $PO_2$ and C-O vibrational bands, respectively (Figure 2d) (Greaves *et al.*, 2023). The sub-diffraction IR absorption of these cells revealed detailed subcellular structures; in the nuclear area, the nuclear membrane and nucleolus were prominent, and in contrast, dense fibrillar components within the nucleoli were characterized by low amide absorption. Furthermore, protein and nucleic acid absorption imaging at 1667 $cm^{-1}$ revealed mitochondria and the 100-nm-sized cisternae structure of the endoplasmic reticulum. The observed binucleation with the fibrillar centers and components is a common phenotype for myeloma cells. Thus, the chemical content on nanometer scale obtained by sSNOM can be used in studying the mechanisms that drive the development and progression of diseases that are manifested through morphological and chemical changes on a sub- and interorganelle level. These studies emphasize IR nanoscopy as a mature technology uniquely suited for direct, label-free, subcellular bioimaging.

## Nanoscale IR monitoring of cellular processes

IR imaging can be leveraged to study metabolic activity and nutrient uptake, and the single-cell capabilities of this technique allow for fine-scale monitoring beyond community-level analyses. A commonly employed technique in this context is stable isotope probing (SIP), which can provide highly detailed insights into cellular function (Alcolombri *et al.*, 2022). During growth, cells can be cultured in the presence of distinct stable isotopes (commonly $^{15}N$, $^{2}H$, or $^{13}C$), which when metabolized are integrated into the molecular building blocks of the cells. As molecular vibrations are inherently affected by variations in atomic mass, isotopic labeling can be detected as characteristic peak shifts of the respective IR bands via the vibrational isotope effect (Figure 3a). Spectroscopic quantification of cellular isotope ratios can thus provide information on cellular metabolism and replication.

Detection of SIP-labeled cells was pioneered using Raman spectroscopy (Wang *et al.*, 2016) and mass spectrometry (i.e., nanoscale secondary ion mass spectrometry, nanoSIMS) (Musat *et al.*, 2016). Recently, IR spectroscopy has been performed on individual SIP-labeled *Escherichia coli* bacteria, being detected by both optical photothermal IR spectroscopy (Lima *et al.*, 2021) and nanoFTIR (Burr *et al.*, 2024). These studies

examined isotope-induced shifts in the amide region, with the former monitoring cellular uptake of $^{13}C$-glucose and $^{15}N$-ammonium chloride, while the latter study focused on the quantitation of single-cell $^{12}C{:}^{13}C$ ratios (Figure 3a). In addition to the proportional $^{13}C$-induced amide I peak shift (from 1655 to 1613 $cm^{-1}$), the spectra were distinctly influenced by isotopic mode coupling, based on the distribution of isotopes within cellular proteins. This resulted in discernible amide I peak shapes, which could be attributed to cellular 'age' (i.e., cells harvested in early- or late-exponential growth phase). Such spectroscopic tracing of metabolic activity at the single-cell level provides insights into intercellular heterogeneity and microbial ecophysiological mechanisms. However, while isotopic labeling provides distinct IR spectral shifts and is a powerful tool to trace biomolecules, its application is limited to systems that can be cultured or experimentally manipulated. For tissue samples, patient-derived material, or complex organisms, isotopic labeling is often impractical or impossible, and IR nanoscopy of these samples must rely on endogenous molecular contrast.

IR imaging has also been utilized for the visualization of nanoparticles and their interaction with cells. Nanoparticles have attracted significant interest in the biomedical field due to their applications as drug delivery systems and in bioimaging (Doane and Burda, 2012), however, their potential toxicological impact also requires careful consideration and assessment (Elsaesser and Howard, 2012). Quantifying the implementation of nanoparticles requires spatial tracking (through the cellular membrane to the intended target location), and the monitoring of their chemical activity and transformation upon biological interaction. Identification and tracking of nanoparticle uptake has been demonstrated using conventional EM-based methods (Elsaesser *et al.*, 2010, 2011), however, for nanoparticles composed of low atomic number materials, EM struggles due to the lack of atomic contrast. An additional challenge is the formation of a so-called protein corona; a dynamic layer of biomolecules that adsorbs onto the nanoparticle surface in biological environments and effectively masks its physicochemical identity (Cedervall *et al.*, 2007). This phenomenon plays a critical role in governing how nanoparticles are recognized and processed by cells. Chemical imaging has long been suggested as a method to track nanoparticle uptake at the single-cell level (Elsaesser *et al.*, 2011). IR nanoscopy now enables direct visualization of not only nanoparticle distributions and chemical transformations, but also the protein corona itself, offering spatially resolved chemical insights into this critical bio–nano interface.

sSNOM imaging has been used to observe the interaction of Au nanoparticles (AuNP) within thin sections of hippocampal neurons (Greaves et al., 2024a). In addition to identification of organelles such as the nuclei, mitochondria, plasma membranes, and the endoplasmic reticulum, this study used sSNOM phase and amplitude imaging to identify AuNP interactions with the neurons in three distinct scenarios: surface association, membrane embedded, and nanoparticle internalization (Figure 3b). Complementary studies have used sSNOM to image the cell volume and morphological features (such as lamellipodia and filopodia) of large (~50–100 μm) malignant glioma cells, and the internalized distribution of mesoporous silica nanoparticles (MSNP; Figure 3c) (Greaves et al., 2024b). MSNP have a wide application in nanomedicine due to their broad biocompatibility, high drug-load capacity, and ease of functional modification. Topographic mapping using the unique absorption of MSNP (at 1100 $cm^{-1}$) within whole glioblastoma cells demonstrates

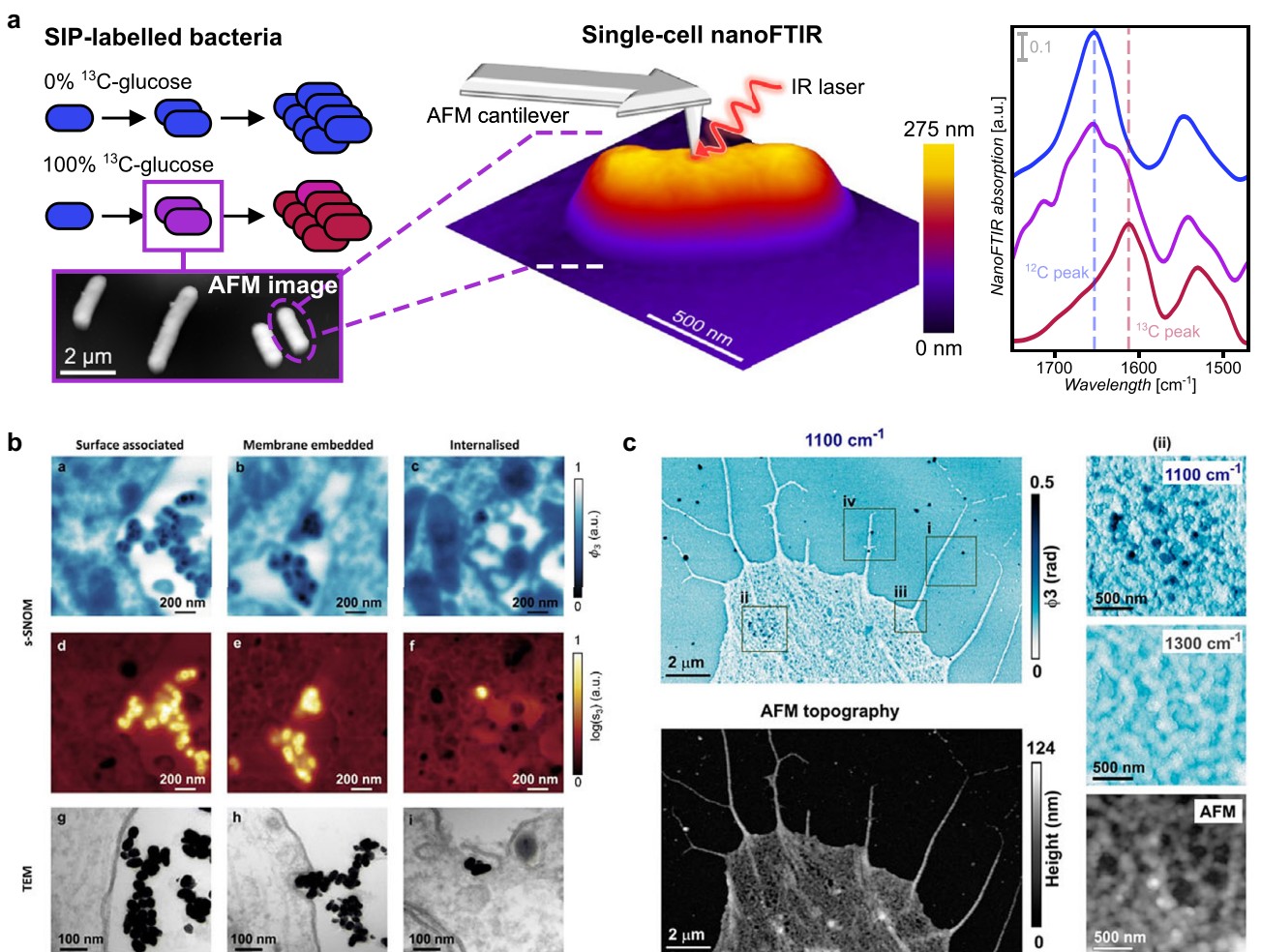

**Figure 3. Applications of IR imaging for monitoring cellular processes.** A. SIP-nanoFTIR as a means of deriving single-cell glucose uptake in *E. coli*. B. Correlative sSNOM absorption, sSNOM reflection (acquired at 1740 cm$^{-1}$), and TEM imaging of AuNP interaction with hippocampal neurons. C. MSNP-treated malignant glioma cell imaged with sSNOM and AFM. Inset (ii) shows the MSNP-associated region, with nanoparticles visible at 1100 cm$^{-1}$, but not at 1300 cm$^{-1}$ nor AFM topography. Figure adapted from (Burr et al., 2024; Greaves, Allison, et al., 2024a) and (Greaves, Pinna, et al., 2024b).

IR-nanoscopic imaging as a nondestructive and chemically informative method for quantifying subcellular nanoparticle distributions. This is crucial for evaluating efficacy in drug delivery and risks of nanotoxicology.

## Emerging technical advances

Recent innovations in analytical instrumentation are enabling opportunities for IR nanoscopy in cellular and microbiological research. Preparing flat samples on reflective substrates now enable the integration of IR nanoscopy with multiple spectroscopic and imaging techniques. This creates opportunities for correlative spectroimaging workflows that greatly expand the breadth of chemical information derived from a single cell. For instance, correlative fluorescence in situ hybridization (FISH)–Raman–SEM-NanoSIMS has been performed on individual SIP-labeled microbial community samples (Schaible *et al.*, 2022). Expanding multimodal techniques into the IR range, however, is still a new concept. Recent examples have focused on complementary photothermal IR methods, such as mid-IR photothermal-FISH (Bai *et al.*, 2023) and optical photothermal IR Raman (Korona *et al.*, 2025). As such, correlative IR nanoscopy is an exciting area of focus for future research.

Conventional IR nanoscopy systems are typically optimized for measurements in dry conditions. The advantage of operating in dry conditions is omitting water absorption contributions that overlaps with signals of biomolecules, a prevalent challenge in IR spectroscopy. However, many applications necessitate nanoscopy in aqueous conditions and recent technical developments are aimed at measurements in liquid; the native environment for many biological samples. Unwanted contributions from water absorption do pose a considerable challenge in bulk IR spectroscopic measurements where the majority of the volume occupied the solvent. Fortunately, in IR nanoscopy this issue is largely omitted as zepto-liter volumes (20 nm × 20 nm × 60 nm ≈ 24·10$^{-21}$ L) of the material directly beneath the AFM tip are probed by the optical near field. Instead, a more considerable issue is the conventional top/side tip illumination geometry in IR nanoscopy, which requires the IR radiation to pass the bulk liquid. To overcome this, liquid sSNOM imaging has been performed on biomembranes and biomimetic peptoid sheets by immersing the AFM tip in an aqueous solution and focus the IR light via bottom illumination using an immersion lens (Pfitzner and Heberle, 2020) or by illuminating the tip with an evanescent field generated by total internal reflection (O'Callahan *et al.*, 2020). Alternatively, living cells have been investigated

utilizing an invertible sample container filled with an aqueous solution covered with an IR transparent membrane. Live micro-organisms can be adhered to a 10-nm-thick SiN membrane and after sample deposition, sSNOM imaging and/or nanoFTIR spectroscopy can be performed through the membrane, using the commonly used side-illumination geometry. In this configuration, the AFM tip is kept in dry conditions, which is characterized by better signal to noise, as compared to when the tip is immersed in liquid. This novel technology has been used to observe cell division and migration of both prokaryotic *E. coli* cells and eukaryotic lung carcinoma cells at a spatial resolution of ~150 nm (Kaltenecker *et al.*, 2021). In addition to visualization over several hours, nanoF-TIR spectra collected from the cell surface (characterized by lipid, protein, and tyrosine peaks) were chemically distinct from the extracellular fluid regions.

While many IR nanoscopy applications have used mid-IR, the spectral range can be expanded through the use of other sources. Examples include THz sSNOM (Hu *et al.*, 2024) or the broadband highly brilliant IR radiation generated by a synchrotron (Bechtel *et al.*, 2020). Synchrotron IR nanospectroscopy (SINS) maintains the spatial resolution of nanoFTIR, while allowing for continuous spectral acquisition from 600 to 4000 cm$^{-1}$, making this output comparable to conventional FTIR spectra. However, as the integrated power over this range is only ~1 mW, single-wavelength imaging is not feasible, and thus it is only possible to perform IR imaging via white light (translation of the interferogram center-burst intensity). While SINS is well established in solid-state physics, applications in biophysics are still in their infancy. However, a recent example of SINS demonstrated protein and lipid changes induced by chemotherapy on whole cancer cells (de Carvalho *et al.*, 2024). This is the first example of a nanoFTIR spectrum of the full fingerprint region of a human cancer cell.

### Enhanced sampling and postprocessing algorithms

A key challenge in nanoscopy is the balance between spatial resolution, scan area, and sufficient signal-to-noise ratio. Biological samples, in particular, have inherently weak IR signals and involve large scan areas, if, for instance, the distribution of biomolecules within a cell is of interest. Weak signals require prolonged pixel integration times to provide sufficient contrast on weak absorption features. For instance, while recording of a single nanoFTIR spectrum acquisition takes a few seconds, long integration times and collecting tens to hundreds of co-additions to obtain a satisfactory signal-to-noise ratio results in total acquisition times in the order of minutes – thus comparable with conventional FTIR spectroscopy. The measurement time for hyperspectral nanoFTIR imaging of micron size areas can therefore be in the order of hours. The scanning speed for sSNOM and AFM-IR is equivalent to AFM scanning, which depending on the scan rate (commonly used 0.5-1 Hz/line), scan area (for cells 5–20 μm), and number of pixels (e.g., $256 \times 256$ px) is on the order of minutes. Recently, machine learning (ML) and enhanced sampling algorithms have begun to address this challenge by enhancing image quality, improving throughput, and reducing acquisition times without sacrificing essential spatial or molecular information.

Prior to the use of ML in sSNOM imaging, various algorithms have been developed for the denoising, image reconstruction, and resolution enhancement in AFM. Such approaches involved, for instance, common methods like wavelet transform based on the nonstationary character of images (Carmichael *et al.*, 2004; Kiwilszo *et al.*, 2012; Schimmack and Mercorelli, 2018) or

deconvolution by the tip geometry (Markiewicz and Goh, 1994; Udpa *et al.*, 2006). However, parameters in these algorithms are user-defined, and as such, genuine objects in an AFM image may be removed or artefacts introduced. Such issues can be overcome by the use of ML, as demonstrated for a range of scanning probe-based techniques (Rahman Laskar and Celano, 2023).

Generative adversarial neural networks (GANs) have been effective across a wide range of scientific applications where denoising or feature enhancement is required (Li, 2023). GANs, initially developed for generating synthetic images (Isola *et al.*, 2017), have found broad utility across various scientific disciplines due to their powerful model-free capabilities (Ahmad *et al.*, 2022). From initial implementations in biomedical imaging to astronomy and particle physics, GANs are increasingly used to enhance quality, fill gaps in incomplete datasets, and simulate complex systems where traditional modeling approaches are insufficient or computationally intensive (Dash *et al.*, 2023). However, traditional GAN approaches require paired image sets, which are difficult to obtain in sSNOM, due to sample drift. Cycle-consistent GANs (CycleGAN) sidestep this issue as the models are trained on unpaired datasets that do not require the same scan areas, or even the same samples (Zhu *et al.*, 2017).

Recently, CycleGANs have been applied to sSNOM images of *C. reinhardtii* cell slices, achieving a denoising performance equivalent to a fourfold increase in pixel integration time (Baiz *et al.*, 2025). The nm-scale absorption features from cellular proteins were significantly enhanced (Figure 4). Furthermore, the method operates on arbitrary image sizes via a patch-based reconstruction technique, preserving detailed spatial information on a subimage level, while enabling larger area scans.

Sparse-sampling techniques are also being adapted from AFM into other probe microscopy methods for optimizing scans, in a way to maximize the information contained in each pixel (Das *et al.*, 2019; Fu *et al.*, 2024). Nonraster scanning and sparse-sampling approaches combined with ML-driven reconstruction algorithms facilitate image reconstruction from significantly reduced data points, further enhancing acquisition capabilities and reducing the data acquisition bottleneck (Liu *et al.*, 2019). These methods exploit spatial and spectral correlations, decreasing overall scanning time. Such ML-based analyses of sSNOM images could further benefit from utilizing orthogonal AFM-based recording modes that can be operated simultaneously. For instance, sSNOM images can be subject to artefacts due to varying tip–sample interaction, because of a heterogeneous surface potential. Combining sSNOM with Kelvin probe force microscopy (where the AFM tip voltage is biased during a measurement) enabled compensation for such undesired effects (Nörenberg *et al.*, 2021); however, such orthogonal data have not been subjected to ML-based analyses to our knowledge.

Hyperspectral IR nanoscopy combines aspects of both nanoF-TIR and sSNOM, capturing full spectral information across multiple spatial locations. Although in practice, hyperspectral acquisition carries challenges related to sample drift, maintaining detector cooling, and AFM probe integrity, given the long data acquisition times (i.e., in the order of several hours, depending on the area of interest; Amenabar *et al.*, 2017). Additionally, hyperspectral imaging yields inherently complex, high-dimensional datasets, making manual analysis challenging. However, automation and ML offer promising means to improve both hyperspectral acquisition and analysis. Autonomous adaptive data acquisition has been applied to scanning microscopy (Holman *et al.*, 2023), and thus could be employed in hyperspectral IR nanoscopy. Similarly,

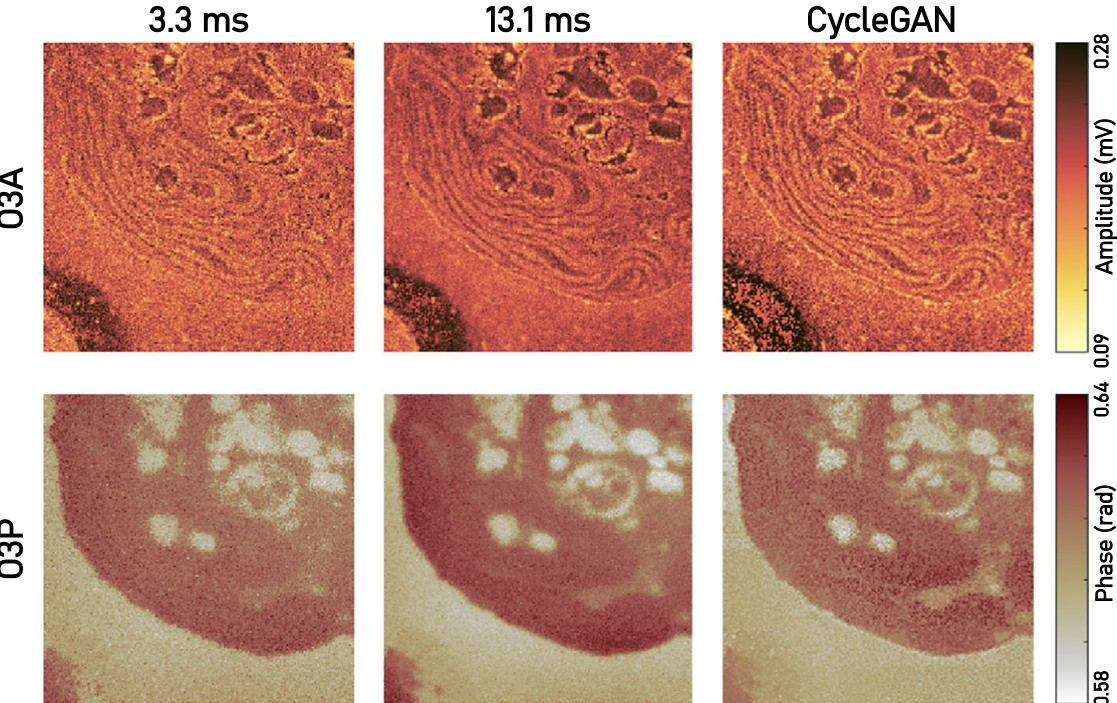

**Figure 4. Example machine learning processing of cell slices**. The same cell is scanned with two different pixel integration times (left, center columns). The noisy image is then processed using a CycleGAN ML model (right column). The optical signals correspond to imaging at 1655 cm$^{-1}$ in resonance with the amide I mode of proteins. Figure reprinted from Baiz, C. R., Kanevche, K., Kozuch, J., & Heberle, J. (2025) *The Journal of Chemical Physics*, 162(5), with the permission of AIP Publishing.

deep-learning models can greatly improve hyperspectral data acquisition (Gao *et al.*, 2024; Keogan *et al.*, 2025), and when combined with ML denoising algorithms (Baiz *et al.*, 2025), the increased signal-to-noise ratio can consequently reduce data acquisition times. Furthermore, the ability of ML algorithms to recognize patterns, highlight correlations, and extract meaningful information from large datasets makes them particularly well suited for the analysis of the rich chemical datasets generated by hyperspectral IR nanoscopy (Chen *et al.*, 2021, 2023). Specifically, supervised learning techniques have been successfully employed for feature identification to facilitate spectral decomposition and have been shown to assist classification of microorganisms (Xue *et al.*, 2024).

The potential for integration of ML algorithms into sSNOM extends beyond mere noise reduction and accelerated imaging. ML can enable automated, hyperspectral analysis, transforming large datasets into biochemical insights. Similar to developments in far-field IR microscopy, the combination of advanced computation and data-rich high-resolution imaging further holds promise for biomedical applications, such as 'digital staining' for subcellular diagnostics (Amrania *et al.*, 2016; Bhargava, 2023).

## Outlook

The studies discussed in this perspective emphasize that IR nanoscopy is a mature technology, with a vast potential for future applications. With this outlook, we offer our insights on how novel instrumentation development, in combination with emerging technologies, can be a path forward to both biophysics and biomedical breakthroughs.

We envision a major focus will be the chemical imaging of subcellular components and single cells in aqueous conditions, utilizing invertible liquid containers (Kaltenecker *et al.*, 2021) or liquid sSNOM setups (O'Callahan *et al.*, 2020; Pfitzner and Heberle, 2020). IR nanoscopy does not yet have the capability to capture the temporal dynamics of biomolecular systems. This is primarily due to the incompatibility of current IR nanoscopy instrumentation with high-speed AFM (HS-AFM). In HS-AFM, the scanning speed is sufficiently fast to spatially resolve dynamic processes in real-time, on millisecond to second timescales (Ando *et al.*, 2013). Therefore, developing an sSNOM system based on HS-AFM technology would provide a way to perform in vivo IR nanoscopic observations. An alternative approach to spatially visualize molecular dynamics is via cryogenically preserving samples at specific time points after the induction of a metabolic process, then analyzing by IR chemical imaging. Cryogenic ultramicrotomy can also yield high-quality cellular thin sections while avoiding invasive resin embedding and thus preserving the native chemical composition.

A route to accessing dynamics on sub-picosecond timescales is via ultrafast nanospectroscopy, an approach combining scanning probe microscopy with pump-probe spectroscopic methods (Jiang *et al.*, 2019). To date, applications of ultrafast nanoscopy are primarily focused on 2D materials, as summarized in the recent review (Zhao *et al.*, 2025). Similarly, pump-probe schemes employed in 2D spectroscopy (Xie *et al.*, 2024; Xie and Xu, 2025) offer information on the time evolution of vibrational bands and their correlation could be adapted to nanoscopy. Implementing this technology to address molecular dynamics on ultrafast timescales would provide unprecedented insight into the innerworkings of cellular building blocks.

The simultaneous multimodal data acquisition of AFM and IR nanoscopy allows sample topography to be recorded alongside chemical content. This is highly applicable for the characterization of biological samples, and can be extended to complementary techniques such as nanoscopy in the visible and THz spectral region with

Kelvin probe force microscopy (Jakob *et al.*, 2021) in order to capture the surface potential of the sample or via photo-induced force microscopy (Shcherbakov *et al.*, 2025) to access the dielectric properties of the sample. When applied to medical applications, such as tissue imaging, IR nanoscopy could be used correlatively with laser-based IR microscopy, allowing for rapid chemical imaging of a large field of view, followed by targeted IR-spectral analysis to resolve the chemical content on inter- and subcellular levels.

Due to the nondestructive nature of this technique, the broad range of compatible biological samples, the potential for correlative studies, and the unique capability to simultaneously derive morphological and chemical information, IR nanoscopic chemical imaging is a particularly exciting technology. The ongoing use of IR nanoscopy continues to advance the fields of biophysics and subcellular imaging.

**Open peer review.** To view the open peer review materials for this article, please visit http://doi.org/10.1017/qrd.2025.10014.

**Data availability statement.** No new data were generated for this perspective.

**Author contribution.** J.H. conceived the article. K.K., D.B., J.D., J.K., C.B., and A.E. wrote the draft. All authors discussed and edited the manuscript.

**Financial support.** J.H. acknowledges funding from the German Research Foundation (DFG, project HE 2063/5-1, and from the SFB 1349 'Fluor-Specific Interactions: Fundamentals and Functions', project number 387284271 – project C05). J.H. and J.K. acknowledge the support from the SupraFAB research building realized with funds from the federal government and the state of Berlin. J.K. acknowledges support from the DFG through the Individual Research Grant (Project No. 500707750). A.E. gratefully acknowledges financial support from the Volkswagen Foundation Freigeist Program and from the Bundesministerium für Wirtschaft und Energie (Projektträger Deutsches Zentrum für Luft- und Raumfahrt) grant numbers 50WB2023 and 50WB2323. C.B. acknowledges Funding from the National Institutes of Health (R35GM133359) as well as the Welch Foundation (F-1881) and a Fellowship from the Alexander von Humboldt Foundation.

**Competing interests.** The authors declare none.

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
