## [Reviewer Report]

In this perspective, Kanevche et al. introduce IR nanoscopy, recent applications of the technology, and future opportunities. Overall, this is a well-balanced review that will educate biologists and vibrational spectroscopists outside of the field of IR nanoscopy and inspire them to enter the field. I particularly appreciate the discussions on how to prepare samples for imaging as this is a critical part of working with these technologies. The IR imaging field is rapidly developing; hence, it is timely to gain a perspective from several of the leaders in the field. I therefore recommend this perspective for publication.

I have a few minor suggestions:

1. A major advantage of IR nanoscopy over other vibrational approaches (and even many fluorescence approaches) is the high spatial resolution (~20 nm). It is unclear that many of the applications discussed required this level of spatial resolution. Why was sNOM the ideal approach to answer the question? I suggest more clearly highlighting where the spatial resolution provided details not available by other approaches or adding additional examples that required the high spatial resolution.

2. While there is a good discussion of strengths and limitations of the technique, two major limitations that could be more clearly addressed are the water background and time to collect images. Water is a problem for most mid-IR approaches. How does sNOM handle this? My understanding is that nanoscopy is quite slow. A transparent discussion of the imaging times would be approached. It may even be appropriate to put these items in context with other approaches – for example the issue of water in sNOM vs other IR approaches, imaging times vs other nanoscale measurements.

3. In the section title “Nanoscale IR Biomarker Monitoring” it was unclear what nanoparticle interactions with the cell had to do with the section title. I suggest creating a more inclusive title (perhaps matching the subtitle of Figure 3) or creating a new subsection.

4. IR shifts are mentioned in the section on “Nanoscale IR Biomarker Monitoring” and the origin of the effect is not explained. I suggest adding a sentence and reference about the shift being related to changes in reduced mass and the harmonic oscillator. This would assist a non-vibrational audience who might not understand the origin of the effect.

5. In a couple of places it is emphasized that nanoscopy is “non-destructive,” however, in the section on live cells it is discussed that live cells can’t be used in contact mode because it will be destructive to the sample. Workarounds are discussed. Therefore, I suggest using relatively non-destructive or acknowledging this limitation more clearly for soft samples.

6. A style suggestion is to capitalize the words in aconyms to distinguish words that make up the acronym.

---

## [Reviewer Report]

In general, the manuscript is very well written, particularly in providing an overview and comparison of several techniques such as s-SNOM, AFM-IR, nano-FTIR, and hyperspectral nano-IR. However, it is not always clear what the differences are between these methods. The review would benefit greatly if the authors could systematically compare the techniques—clarifying what each one measures, their respective limitations, and where they are complementary. For example, while the authors mention that AFM-IR and s-SNOM are complementary, a deeper explanation of this complementarity is missing.

The article would be significantly strengthened by including a comparative table summarizing the main features, advantages, limitations, and application areas of all the techniques discussed. In addition, a schematic or cartoon illustrating the key technical differences would make the review more accessible, especially for readers from the biological sciences. At present, the emphasis of the review seems somewhat weighted toward s-SNOM, and balancing the coverage across techniques would improve the overall scope.

It would also be valuable to more explicitly address the main bottlenecks in advancing biophotonics for biological imaging. These include challenges of data interpretation in complex cellular contexts as well as the practical advantages and disadvantages of operating in contact with biological samples using an AFM tip (e.g., issues of reproducibility, mechanical fragility, and the presence of sticky particles). To date, these methods have been successfully applied to isotope-labeled IR tags, but this is not always feasible, and such limitations should be acknowledged. In particular, it remains unclear to what extent these techniques can be applied to tissue samples and how nano-IR approaches can be combined with other modalities to enable true multimodal analysis. A discussion of these aspects would provide readers with a more realistic perspective on the current capabilities and future challenges of IR nanoscopy in biology.

---

## [Reviewer Report]

Please check and correct if needed:

In Figures 1 and 2, the AFM topography of a cells is shown, but the corresponding IR image appears to represent most likepy a resin-embedded cell that was cut into an ultrathin section (most likely TEM-thin). In my understanding, in that case, the topography should appear relatively flat.

---

## [Editor Report]

One Reviewer had a comment that Authors may wish to consider:

In Figures 1 and 2, the AFM topography of a cells is shown, but the corresponding IR image appears to represent most likepy a resin-embedded cell that was cut into an ultrathin section (most likely TEM-thin). In my understanding, in that case, the topography should appear relatively flat.